# Photoaging and Sequential Function Reversal with Cellular-Resolution Optical Coherence Tomography in a Nude Mice Model

**DOI:** 10.3390/ijms23137009

**Published:** 2022-06-23

**Authors:** Yen-Jen Wang, Chang-Cheng Chang, Meng-En Lu, Yu-Hung Wu, Jia-Wei Shen, Hsiu-Mei Chiang, Bor-Shyh Lin

**Affiliations:** 1Department of Dermatology, MacKay Memorial Hospital, Taipei 10449, Taiwan; yenjen4208@gmail.com (Y.-J.W.); yuhung_wu@yahoo.com (Y.-H.W.); 2Department of Cosmetic Applications and Management, MacKay Junior College of Medicine, Nursing, and Management, New Taipei City 25245, Taiwan; 3Department of Cosmeceutics, China Medical University, Taichung 40433, Taiwan; lemon520627@gmail.com (M.-E.L.); winnie953444@yahoo.com.tw (J.-W.S.); hmchiang@mail.cmu.edu.tw (H.-M.C.); 4Institute of Imaging and Biomedical Photonics, National Yang Ming Chiao Tung University, Tainan 71150, Taiwan; borshyhlin@nctu.edu.tw; 5School of Medicine, College of Medicine, China Medical University Hospital, China Medical University, Taichung 404332, Taiwan; 6Aesthetic Medical Center, China Medical University Hospital, Taichung 40402, Taiwan; 7Department of Medicine, Mackay Medical College, New Taipei City 25245, Taiwan

**Keywords:** nude mice, optical coherence tomography, ultraviolet, photoaging reversal, laser-induced optical breakdown

## Abstract

Although nude mice are an ideal photoaging research model, skin biopsies result in inflammation and are rarely performed at baseline. Meanwhile, studies on antiphotoaging antioxidants or rejuvenation techniques often neglect the spontaneous reversal capacity. Full-field optical coherence tomography (FFOCT) can acquire cellular details noninvasively. This study aimed to establish a photoaging and sequential function reversal nude mice model assisted by an in vivo cellular resolution FFOCT system. We investigated whether a picosecond alexandrite laser (PAL) with a diffractive lens array (DLA) accelerated the reversal. In the sequential noninvasive assessment using FFOCT, a spectrophotometer, and DermaLab Combo^®^, the photodamage percentage recovery plot demonstrated the spontaneous recovery capacity of the affected skin by UVB-induced transepidermal water loss and UVA-induced epidermis thickening. A PAL with DLA not only accelerated skin barrier regeneration with epidermal polarity, but also increased dermal neocollagenesis, whereas the nonlasered group still had >60% collagen intensity loss and 40% erythema from photodamage. Our study demonstrated that FFOCT images accurately resemble the living tissue. The photoaging and sequential function reversal model provides a reference to assess the spontaneous recovery capacity of nude mice from photodamage. This model can be utilized to evaluate the sequential noninvasive photodamage and reversal effects after other interventions.

## 1. Introduction

Photoaging was considered irreversible for centuries until Kligman et al. found it partially reversible in hairless mice [1]. Since then, efforts have been undertaken to reverse photoaging using retinoic acid and topical/systemic natural antioxidants [2,3]. Sub-nanosecond and picosecond laser technologies have further revolutionized the field of photodamage repair [4,5]. Laser-induced optical breakdown (LIOB) creates microscopic vacuoles at the sites of optical breakdown and stimulates an epidermal repair mechanism with neocollagenesis [5,6].

The assessment of therapeutic outcomes for photoaging reversal requires faithful preclinical models. The first photoaged hairless animal model was established in 1996, after an attempt to use a haired mouse in 1964 [7]. Hairless mice, such as Skh-1 and Skh-2, contain numerous keratinizing cysts that enlarge with thin walls, rupture, and provoke a granulomatous reaction as aging [8]. Nude mice lack trans-urocanic acid and melanin to absorb UV irradiation and have an incompletely formed stratum corneum with poorly aligned keratin filaments, allowing efficient UV radiation penetration [9]. Therefore, nude mice undergo accelerated photoaging and have become an ideal photoaging animal model [10,11].

Previous nude mice studies for photoaging reversal used digital photographs for dorsal wrinkle evaluation and skin sampling for histological observations [10,12]. However, a biopsy results in inflammation and interferes with the interpretation of UV-induced photoinflammation. Furthermore, biopsies can only assess a focal area of the involved skin and are seldom performed at baseline. Photodamage involves all compartments of the skin and encompasses both acute/chronic inflammation and dermal barrier dysfunction. Therefore, a noninvasive sequential method for a whole-skin evaluation is warranted to avoid biopsy-related sampling errors.

Optical coherence tomography (OCT) provides cross-sectional images similar to a hematoxylin and eosin (H&E) staining-based histological examination, but was initially utilized only in skin cancers due to its limited resolution [13,14,15]. By utilizing a high-brightness broadband-extended source with a reliable speckle suppression method, the full-field OCT (FFOCT) system can deliver a cellular resolution and “noninvasive digital skin biopsy” [13]. Therefore, this study aimed to establish a photoaging and sequential function reversal model in nude mice by a noninvasive evaluation using DermaLab Combo^®^, a spectrophotometer, and sequential real-time cellular-resolution FFOCT images.

## 2. Results

### 2.1. The Established Nude Mice Photoaging Reversal Model with Photodamage Percentage

Through sequential noninvasive assessment using the cellular-resolution FFOCT, a spectrophotometer, and DermaLab Combo^®^ at different points of time, we calculated the post-induction recovery ratio (%) of TEWL, erythema index, and collagen intensity in the UVB-L(−)/UVB-L(+) and UVA-L(-)/UVA-L(+) groups using the formulae mentioned below, and the photodamage percentage recovery plot was depicted accordingly.

Post induction recovery ratio (%) of UVB-L(−) and UVB-L(+) group
$=\frac{\left|f_{12}^{14}\left(n\right)=w_{n}-w_{11}\right|}{\left|w_{11}-w_{0}\right|}$ ∗ 100%
Post induction recovery ratio (%) of UVA-L(−) and UVA-L(+) group
$=\frac{\left|f_{12}^{14}\left(n\right)=w_{n}-w_{11}\right|}{\left|w_{11}-w_{2}\right|}$ ∗ 100%
Photodamage percentage (%) = (1 − post UV induction recovery ratio) × 100%**W0**: baselines of UVB irradiated mice**W2**: baselines of UVA irradiated mice**W10**: complete UVA or UVB irradiation.**W11**: peak of photodamage**UVB-L(****−) and UVB-L(+) group**: UVB irradiation for 10 weeks (W0–W10)**UVA-L(****−) and UVA-L(+) group**: UVA irradiation for 8 weeks (W2–W10)**Quantitative Parameters of photodamage**: erythema, collagen intensity, TEWL, epidermal thickness (OCT).W10: complete UV irradiation in both groups

Here, 100% represents the maximum photodamage post UV induction, while zero represents a reversal of photoaging with minimal damage; values below 0 indicate rejuvenation. The baseline parameters of the UVA- and UVB-irradiated mice were measured at W2 and W0, respectively. All irradiated mice completed irradiation at W10 and were then allowed to undergo either spontaneous reversal (UVA-L(−) and UVB-L(−)) or laser-assisted reversal (UVA-L(+) and UVB-L(+)). The maximal photodamage (100%) was at W11 because the photodamage persisted beyond one week after induction. Spontaneous or laser-accelerated reversal ensued from W12, and the photodamage gradually decreased (Figure 1).

### 2.2. Nude Mice Skin FFOCT Characteristics

#### 2.2.1. Stratum Corneum and Epidermis

Epidermal and hair keratinization is genetically impaired in nude mice [9,16]. The defective stratum corneum forms an irregular and discontinuous thin uppermost hyper-reflective layer. It is sometimes associated with a gray zone and a thin layer of hyper-reflective cells, which together constitute the “sandwich sign” (Figure 2a). This thin layer of hyper-reflective cells is the stratum granulosum with more keratohyalin granules than the haired layer [9]. Keratinocyte nuclei appear as oval, hypo-reflective structures in the epidermis. Nude mice do not produce melanin; therefore, the basal layer is not hyperreflective. The dermal–epidermal junction (DEJ) was clearly visible. In vivo FFOCT additionally provided the epidermal thickness values. Epidermal thickness measured with OCT was comparable with that ascertained by hematoxylin and eosin (H&E) staining-based measurements (Appendix A).

#### 2.2.2. Dermis

The epidermis and dermis were clearly separated by the dermal–epidermal junction. The blood vessels and parallelly arranged, hyper-reflective collagen bundles could be identified. Hair bulbs were embedded in the hypodermis, but short crippled hair shafts seldom emerged from the skin surface (Figure 2c).

### 2.3. UV-Induced Epidermal Thickening and Nuclei Disorientation Observed by FFOCT

Epidermal thickness significantly increased following UV irradiation (Figure 2, Figure 3 and Figure 4a,b). The UVB irradiated groups demonstrated a thickened epidermis with epidermal hypergranulosis and hyporeflective nuclei irregular in size, shape, and orientation (loss of polarity) compared to that in the UVA irradiation group (Figure 3b,h). No obvious elastosis was observed after UVB and UVA irradiation in nude mice. The control mice did not show noticeable skin changes over time (Appendix A).

### 2.4. Limited Photoaging Reversal Capacity from Spontaneity

After induction with UVA/UVB, the increased scratching by the mice resulted in excoriation wounds that usually resolved within 2–3 days. The recovery ratios of collagen intensity loss, erythema index, transepidermal water loss (TEWL), and epidermal thickening in UVA-L(−)/UVB-L(−) mice at the end of the experiment (4 weeks after induction) were 28.4 ± 27.1%/35.1 ± 31.4%, 61.4 ± 22.6%/51.3 ± 35.7%, 45.6 ± 44.3%/94.4 ± 97.5%, and 114.8 ± 95.0%/25.2 ± 17.6%, respectively (Figure 5).

### 2.5. LIOB Generation Post-Picosecond Laser with Diffractive Lens Array in Nude Mice Despite Lack of Melanin

The LIOB was traced using sequential OCT images immediately and 120 h after the initial laser therapy in both the UVA-L(+) and UVB-L(+) groups (Figure 6). The LIOB appeared as a round-to-oval intraepidermal hyporeflective vacuole on FFOCT, expanding to one-third to two-thirds of the epidermal thickness. The basement membrane (BM) remained intact. LIOB was still observed in the UVB-L(+) group 3 weeks after the third laser therapy (Figure 6i). The diameter of the LIOB in sequential images was measured, and the mean diameter of the LIOB in the UVB-L(+) group was 35.45 ± 1.74 μm, which was larger than that of the UVA-L(+) group (24.75 ± 5.36 μm). Microscopic epidermal necrotic debris (MENDS) was observed in the post-treatment zones as hyper-reflective stacks situated immediately inferior to and within the stratum corneum (Figure 6d).

### 2.6. Accelerated Epidermal Thickness Recovery and Increased Epidermal Polarity Post Laser Therapy Observed by FFOCT

At the end of the experiment, no considerable differences were observed in the body weights of mice in all three groups (Appendix A). The laser treatment significantly increased epidermal polarity in the UVB-irradiated groups and significantly accelerated epidermal thickening recovery in the UVB induction group (*p* < 0.05). (Figure 3, Figure 4 and Appendix A). At the end of the experiment, both lasered groups demonstrated a decreased epidermal thickening (H&E staining): 36.74 μm in the UVA-L(−) group vs. 29.57 μm in the UVA-L(+) group, *p* < 0.01; 77.62 μm in the UVB-L(−) group vs. 34.05 μm in the UVB-L(+) group, *p* < 0.0001) (Appendix A).

### 2.7. Accelerated Collagen Intensity Recovery and Skin Barrier Restoration Post Laser Therapy

Collagen intensity decreased regardless of UVA or UVB exposure (Appendix A). The UVA-L(+) group regained more collagen at W14 (25.01 ± 2.49 in the UVA-L(+) group vs. 24.02 ± 3.63 in UVA-L(−) group, *p* < 0.05; Figure 4c), whereas no significant difference in collagen intensity was observed in the UVB-L(−) and UVB-L(+) groups at W14 (21.55 ± 3.45 in the UVB-L(+) group vs. 20.68 ± 1.52 in the UVB-L(−) group, *p* > 0.05; Figure 4d). Δ Collagen intensity (difference between W10/W14 and baseline) was significantly lower in the UVA-L(+) group than that in the UVA-L(−) group (Appendix A).

#### 2.7.1. Erythema Index

The a* value, an index of erythema that represents the degree of skin inflammation, gradually increased and peaked at the end of UVA and UVB induction. Erythema gradually decreased during the following four weeks but did not completely recover to the baseline level. The UVA-L(+)/UVB-L(+) groups demonstrated an accelerated recovery from erythema compared with that in the UVA-L(−)/UVB-L(−) groups (Appendix A). (The a* value for erythema was 8.240 in the UVA-L(+) group vs. 8.593 in the UVA-L(−) group (*p* = 0.589) and 7.998 in the UVB-L(+) group vs. 9.125 in the UVB-L(−) group (*p* < 0.05).)

#### 2.7.2. Transepidermal Water Loss (TEWL)

TEWL was slightly altered after UVA irradiation and significantly increased after UVB induction (Figure 4c,d). All UV-irradiated groups still had increased TEWL compared to the control group at the end of the experiment. However, all the lasered groups had lesser TEWL than that in the nonlasered groups: 14.23 g/m^2^h in the UVA-L(+) group vs. 14.96 g/m^2^h in the UVA-L(−) group (*p* = 0.8671) and 27.08 g/m^2^h in the UVB-L(+) group vs. 27.81 g/m^2^h in the UVB-L(−) group (*p* = 0.9860) (Appendix A). Δ TEWL (difference between W10/W14 and baseline) was lower in the UVB-L(+) group than that in the UVB-L(−) group, although the difference was not significant (Appendix A).

### 2.8. Depiction of the Photodamage Percentage Plots between Laser vs. Nonlaserd Groups

Laser therapy significantly accelerated the recovery of nude mice from the UVB-induced collagen intensity loss at W13 and W14 (*p* < 0.05; Figure 5a) and the UVA-induced erythema at W14 (*p* < 0.05; Figure 5b). The photodamage percentage decreased with steeper slopes in the laser group compared with the nonlasered group. There was a lesser degree of photodamage at the end of the experiment in the lasered groups, whereas >60% photodamage of collagen intensity and >40% of erythema were still observed in the nonlasered group (Figure 5a,b). Laser reduced TEWL post UVA induction and epidermal thickening post UVB induction (Figure 5c,d). However, at the end of the experiment, the UVA-L(−) group still showed a photodamage percentage of TEWL of more than 50% (Figure 5c).

## 3. Discussion

In vivo OCT is an emerging privilege for repetitive real-time and touchless evaluation, which preserves the cellular signal transduction and provides the characteristics at baselines in living models [17,18]. The trend of epidermal thickness changes measured with FFOCT in our study corresponded to those determined from H&E staining-based histologic examinations. Without the dewaxing, dehydration, H&E staining, and cover-slipping process, we believe the FFOCT evaluation can vividly and accurately resemble the living model condition. DermaLab Combo^®^ and the spectrophotometer completed the skin barrier function and collagen intensity assessment. Surgery or biopsy results in the recruitment of inflammatory cells to the site of injury, including the influx of neutrophils and monocytes and the release of proinflammatory cytokines such as interleukin (IL)-1b, IL-6, and tumor necrosis factor-α [19]. The proinflammatory effect of a biopsy may interfere with the interpretation of UVR-induced photoinflammation and limit the capacity of reversal. Reflectance confocal microscopy (RCM) is a valid tool for the noninvasive diagnosis and monitoring of superficial skin cancers and pigment disorders [20,21,22]. However, en face images require substantial learning and practice for interpretation. Meanwhile, the BM and epidermal thickness cannot be directly evaluated/measured using RCM [14,15,23]. The sequential evaluation with cellular-resolution FFOCT enabled us to observe the loss of cell polarity after UVB induction and the laser-accelerated recovery of epidermal cell polarity [13]. FFOCT can generate overwhelmingly large data for artificial intelligence (AI), and the AI-assisted quantification system can enhance the data process acuity and efficiency. The advantages of biopsy and FFOCT are listed in Appendix A.

In previous studies, the intraepidermal or dermal vacuole formation created with laser therapy was mainly related to chromophores such as hemoglobin or melanin [24]. We successfully produced vacuoles in the epidermis of melanin-lacking nude mice with an intact BM through 755 nm PAL with DLA, which was clearly demonstrated by cross-sectional images provided by the FFOCT. LIOB is a nonlinear multiphoton absorption process in which the irradiance threshold for breakdown is surmounted and observed by OCT [25]. The threshold energy required to generate seed electrons for LIOB formation exclusively via multiphoton absorption is approximately 10^13^ W/cm^2^ and cannot be safely induced in human skin [26,27]. The thermal initiation pathway assisted by chromophores results in a lower irradiance threshold, which lowers the irradiance threshold by 20 times [27], whereas ubiquitous water lowers the irradiance threshold by 10–100 times [27,28]. Therefore, transparent and weakly absorbing tissues such as the cornea mainly rely on the multiphoton-initiated optical breakdown, whereas highly absorbing tissues such as dark skin require a much lower energy threshold [24,29]. The irradiance energy for nude mouse skin in our study was 0.00306667 × 10^13^ W/cm^2^, which was approximately 362 times lower than that via the multiphoton absorption process (10^13^ W/cm^2^). Therefore, the radiation absorption characteristics of the nude mice epidermis fall between those of the transparent cornea and the human skin.

LIOB initiated neocollagenesis and neoelastinogenesis beyond the depth of acute tissue injury and contributed to accelerated skin barrier regeneration and collagen intensity recovery compared with the nonlasered groups. Damage to dermal resident cells (fibroblasts, macrophages, and mast cells) resulted in the initiation of the inflammatory response [6]. This is a corollary of the keratinocyte–fibroblast interactions in wound healing and barrier functions [30]. Fibroblasts promote epidermal thickening to ameliorate UVB-induced skin damage by inducing the keratinocyte growth factor (KGF) [31,32]. The epidermis also thickens in the UVA-irradiated mice, mediated by the KGF and the nuclear factor erythroid 2-related factor 2 (Nrf2) pathway [33,34,35]. Nude mice lack melanin, and the supranuclear cap of “parasol” melanokerasomes is not formed [36]. Therefore, DNA in the nuclei of keratinocytes becomes the major chromophore in UVB-irradiated mice, resulting in a lower LIOB formation threshold. In contrast, more UVA was absorbed in the dermis, and the size of the LIOB was smaller than that after UVB irradiation. Our study showed that the presence of melanin and hemoglobin is not a prerequisite for LIOB formation [24,29]. To create a breakdown, the irradiance threshold is a function of both medium characteristics, including the ionization energy and impurity level, and laser beam characteristics (wavelength/pulse width/spot size).

Although nude mice demonstrated some capacity for spontaneous recovery in UVB-induced TEWL and UVA-induced epidermal thickening, erythema and collagen intensity incompletely recovered with >40% and >60% photodamage, respectively. Based on a skin barrier function assessment and sequential OCT images, PAL with DLA not only increased epidermal polarity, but also accelerated collagen intensity, epidermal thickening, and skin barrier regeneration.

The photodamage reversal of skin changes after UV irradiation in humans occurs at a much slower pace [37]. Hydration returns to the baseline level after one month (1.5 minimal erythema dose (MED)), whereas the complete recovery of erythema takes six months even with only one MED UV [37]. Testing therapeutic approaches requires living preclinical models, which faithfully simulate the in vivo hallmarks of the photoaging process in situ. Erythema and pigmentation responses are important acute UV radiation inflammatory effects that cannot be assessed in vitro, and frequent biopsies from all age groups in humans are not feasible [38]. The drawbacks of HaCaT cells are their inability to reproduce the complexity of the skin, cellular heterogeneity, and crosstalk between the epidermis and dermis. Animal models with accelerated photoaging processes are desirable [2,39,40]. On the other hand, studies examining the therapeutic ability of compounds or techniques often neglect the spontaneous photoaging reversal capacity after UV insult is halted [2,41,42]. Our study provided a reference for the time duration needed for epidermis and barrier function repair and the degree of spontaneous reversal/laser accelerating reversal from UV damage in nude mice. The photodamage percentage recovery plot depicted by the serial noninvasive assessments and the large numbers of FFOCT images would enable the testing of the photoaging reversal effects of antioxidant compounds, nanofats, stromal vascular fraction, and adipose-derived stem cells.

Our study had some limitations. First, nude mice do not perfectly resemble the changes seen in human skin, but they can be an excellent model for xenografts and understanding the mechanisms of photoaging. Second, full contact and the immobilization of the probe are critical aspects for even more delicate resolution imaging techniques such as the FFOCT, and handheld devices with smaller probes would be desirable. A learning curve for OCT image interpretation and technique improvements is required.

In summary, cellular-resolution FFOCT is a novel research technique for the sequential, delicate, and noninvasive photodamage evaluation, including the assessment of epidermis thickening and epidermal nuclei polarity. Our study provided a reference for the spontaneous photodamage reversal capacity of nude mice and the laser-assisted repair of UV damage. Laser-induced LIOB accelerated the recovery of collagen intensity loss, erythema, TEWL from UVA damage, and epidermal thickening from UVB damage to a minimal degree. In contrast, the nonlasered group demonstrated >60% collagen intensity loss and >40% erythema photodamage. FFOCT images accurately reproduced the structure of living tissue in vivo and are valuable for the touchless evaluation of the therapeutic effects of the intervention.

## 4. Materials and Methods

### 4.1. Experimental Animals

All experimental protocols were approved by the Institutional Animal Use and Care Committee of China Medical University (protocol no. CUMIACUC-2020-330). Five-week-old female BALB/cAnN.Cg-Foxn1^nu^/CrlNarl hairless mice were obtained from BioLASCO, Ltd. (Taipei, Taiwan). The mice were maintained under controlled conditions: temperature, 22 ± 4 °C; relative humidity, 50 ± 10%; light–dark cycle, 12 h.

After 1-week of acclimatization, the nude mice were randomly allocated into five groups:Control group: did not receive UVA or UVB treatment;UVA-L(−) group: UVA-irradiated without laser treatment;UVA-L(+) group: UVA-irradiated followed by laser treatment;UVB-L(−) group: UVB-irradiated without laser treatment;UVB-L(+) group: UVB-irradiated followed by laser treatment.

Each group had eight mice. All treatments were administered to the dorsal skin.

### 4.2. UVB and UVA Irradiation

The UVB and UVA photoaging models were established based on the experimental protocol of Wu and Lan [3,43]. For UVB radiation, UV light (broadband with peak emission at 302 nm, CL-1000 M (Analytik Jena US LLC, formerly UVP, California, Upland, CA 91786, USA) was used. UVB-irradiated mice were exposed to a gradually increasing dose of UVB irradiation thrice a week: first week, 36 mJ/cm^2^; second to fourth week, 54 mJ/cm^2^; fifth to the seventh week, 72 mJ/cm^2^; and eighth to the tenth week, 108 mJ/cm^2^ [3].

UVA-irradiated mice received energy of 8 J/cm^2^, three times a week for 8 weeks with CL1000 L UVA light (UVP, Upland, USA; emission spectrum 320–400 nm with a peak wavelength at 365 nm) [43].

### 4.3. Spontaneous Recovery from UV Photodamage

After complete UV induction, the UVA- and UVB-irradiated groups received no additional procedure and spontaneously recovered from photodamage in the following four weeks. These groups were designated as UVA-L(−) and UVB-L(−), respectively.

### 4.4. Picosecond Laser with Diffractive Lens Array (DLA)

UVA-L(+) and UVB-L(+) mice were treated with a 755 nm ps PAL (PicoSure^®^, Cynosure, Westford, Massachusetts, USA) with a DLA (FOCUS Lens Array) for three days after UVA/UVB induction. The parameter settings were an average fluence of 0.71 J/cm^2^ from a 6 mm spot diameter with 750 ps pulse duration. Each mouse received a total number of counts of 500 passes over the 2.5 cm × 3 cm dorsal back area for three consecutive days after UVA or UVB irradiation. Owing to the DLA redistribution of the energy, the peak fluence of high-energy zones (microbeams, 10% of tissue) was 23 J/cm^2^, and 90% of the tissue in the low-fluence background received only 0.25 J/cm^2^.

### 4.5. Image Acquisition by In Vivo OCT

FFOCT (ApolloVue^®^ S100 Image System; Apollo Medical Optics, Ltd., Taipei, Taiwan) was used for nude mouse skin characteristic measurements, including epidermal thickness and nucleus polarity. This system is a time-domain Mirau-type FFOCT with a high-brightness broadband light source based on a glass-clad Ti^3+^/sapphire crystal fiber with a central wavelength of 780 nm. A high-numerical-aperture objective lens and dynamic focus achieved a high lateral resolution of 1 μm. The axial resolution could reach 1.3 μm on cross-sectional scans, and the scanning depth was 400 μm. This resolution was optimized for the conditions of a human skin model (epidermal thickness, 80–200 μm). The nude mice lacked a well-formed stratum corneum, and epidermal thickness ranged from 30 to 70 um. Therefore, the resolution varied between 1 and 2 μm according to the optical characteristics of the sample, which was still sufficient for most image features. The system provided both cross-sectional and en face imaging modes, and the corresponding fields of view were 500 μm (width) × 400 μm (depth) and 500 μm × 500 μm, respectively.

Sequential images at baseline, immediately post complete UV induction, and four weeks following the postinduction were documented. In the laser groups, the irradiated nude mouse skin was scanned immediately and the first few days post laser irradiation to identify LIOB (Figure 3a).

### 4.6. Measurement of Skin Erythema, Collagen Intensity, and Transepidermal Water Loss (TEWL)

The effects of UVB- and UVA-induced erythema (a* value) were tested every 2 weeks using a spectrophotometer (SCM-104/108, Ruyico Technology Corporation, Taipei, Taiwan) [3]. DermaLab Combo^®^ (Cortex Technology ApS, Hadsund, Denmark) with a 20 MHz focused ultrasound probe was used to assess collagen intensity (AU) and TEWL [44]. The biological meaning of collagen intensity measurements was the density and amounts of collagen based on the reflection by the echo probe.

Biopsies were performed once induction was completed and again after four weeks of spontaneous/laser-assisted recovery.

### 4.7. Preparation of Skin Specimens and H&E Staining

Two mice in the UVB- and UVA-irradiated groups were sacrificed while completing UV induction (W10). Mice in the UVA-L(+) and UVB-L(+) groups received 3.5 mm punch biopsies every week post laser treatment. At W14, the mice were sacrificed for whole dorsal skin histological H&E examination [3]. The epidermis thickness was determined using ImageJ software (National Institutes of Health, Bethesda, MD, USA).

### 4.8. Statistical Analysis

The results were expressed as the mean ± standard deviation. Data between different groups were statistically analyzed using one-way ANOVA, followed by Tukey’s test. *p*-values < 0.05 were considered statistically significant. All analyses were performed using the GraphPad Prism 9 software (GraphPad Software Inc., La Jolla, CA, USA). The photodamage percentile after induction (e.g., at W12, W13, and W14) between the irradiated and control groups was compared using the generalized estimating equation (GEE). The linking function was identified, and the distribution was normal in the GEE. The working correlation matrix was exchangeable, and a robust standard error was adopted. The UVA and UVB groups were analyzed separately. Data analyses were conducted using SPSS version 26 (IBM SPSS Inc., Chicago, IL, USA). A two-sided *p*-value < 0.05 was considered statistically significant.

## Figures and Tables

**Figure 1 ijms-23-07009-f001:**
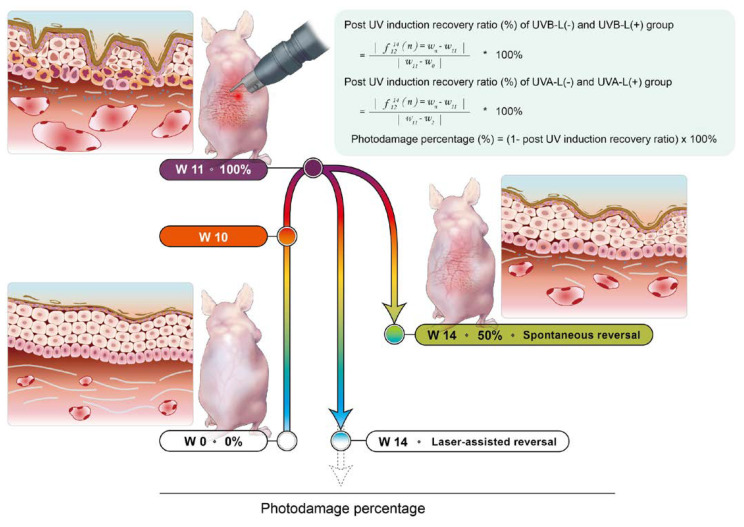
The established nude mouse photoaging reversal model. UV induction was started at baseline (W2 for UVA-irradiated and W0 for UVB-irradiated groups) and was complete at W10. The lasered groups immediately received laser for 3 consecutive days. Photodamage persisted beyond 1 week after induction, and the photodamage reached 100% at W11. This was followed by spontaneous reversal or laser-assisted reversal over the following four weeks. Mice were sacrificed at W14.

**Figure 2 ijms-23-07009-f002:**
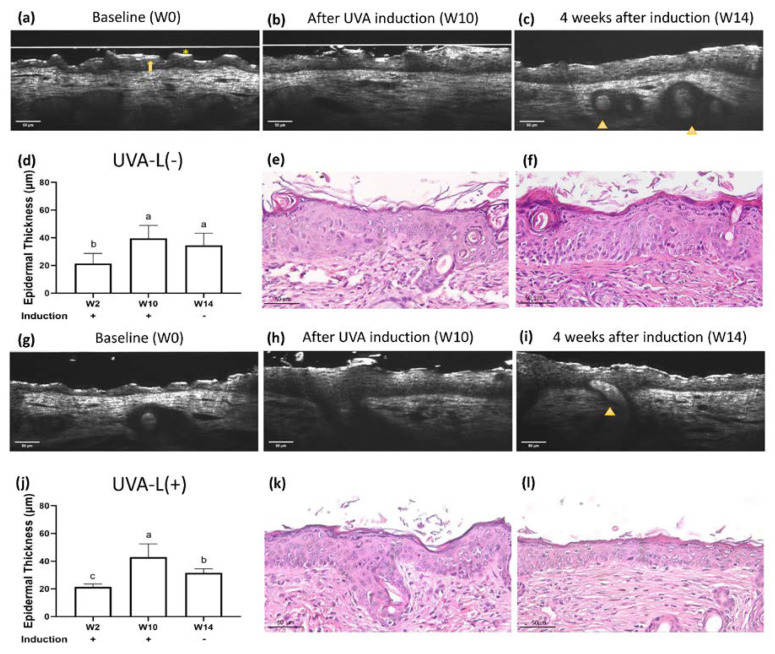
Real-time optical coherence tomography (OCT) and histology images of UVA-L(−) nude mice at (**a**) baseline (W2), (**b**) complete UVA induction (W10), and (**c**) 4 weeks after spontaneous recovery (W14). The UVA-L(+) group received laser immediately after UVA induction. Real-time OCT images of UVA-L(+) nude mice at (**g**) W2, (**h**) W10, and (**i**) W14. Nude mice have a defective stratum corneum, which appears as an irregular and discontinuous thin uppermost hyperreflective layer (*). It is sometimes followed by a lower gray zone and a thin layer of hyperreflective stratum granulosum, together contributing to the “sandwich sign” (arrow). Abortive hair bulbs are embedded in the hypodermis (arrowhead). (**e**,**k**) After UVA irradiation, perinuclear vacuoles in keratinocytes, mild intercellular edema, and epidermal acanthosis were observed. In the dermis, endothelial cells were enlarged, and the perivascular infiltration with mast cells, neutrophils, and lymphocytes was prominent. Epidermal thickness changes in two groups, shown in (**d**,**j**). Different letters indicate statistically significant differences (*p* < 0.05). The UVA-L(+) group showed better recovery from epidermal thickening than the UVA-L(−) group. (**f**,**l**) The laser group (**l**) demonstrated lesser epidermal acanthosis and decreased perivascular inflammatory cells infiltration compared to the nonlasered group (**f**).

**Figure 3 ijms-23-07009-f003:**
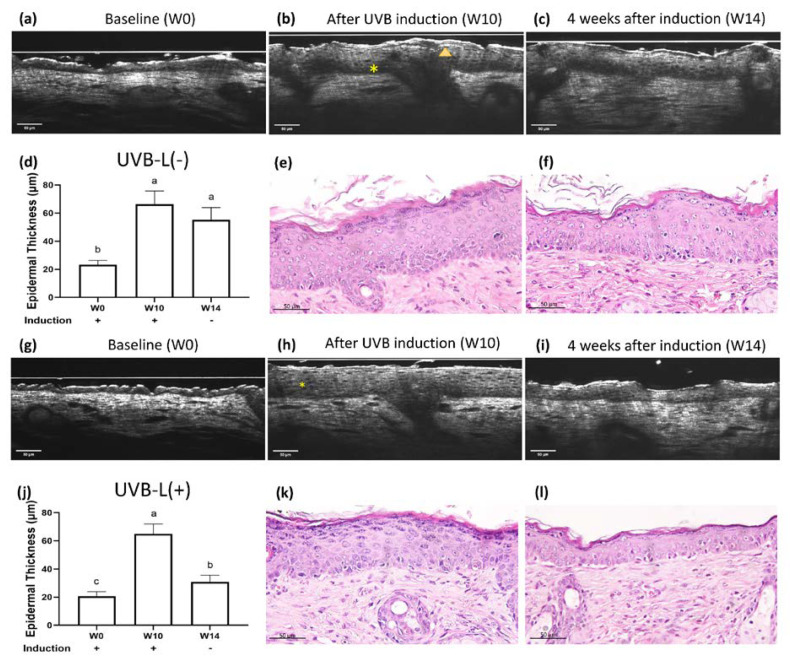
Real-time OCT and histology images of UVB-irradiated nude mice at (**a**) baseline (W0), (**b**) complete UVB induction (W10), and (**c**) 4 weeks after spontaneous recovery (W14). The UVB-L(+) group mice received laser immediately after UVB induction. Real-time OCT images of the UVB-L(+) group nude mice at (**g**) W0, (**h**) W10, and (**i**) W14. UVB irradiation resulted in hypergranulosis (arrowhead) and thickened epidermis with nuclei irregular in size, shape, and orientation (*). (**e**,**k**) After UVB irradiation, significant hypergranulosis, epidermal acanthosis, intercellular edema, occasional apoptotic cells, and mild exocytosis of lymphocytes were observed. Loss of cell polarity with enlarged hyperchromatic nuclei was prominent in the basal layer. (**f**,**l**): The lasered group (**l**) regained cell polarity and demonstrated decreased epidermal thickening/hypergranulosis, while the non-lasered group (**f**) still showed significant hypergranulosis, epidermal acanthosis, and loss of cell polarity. (**d**,**j**) Different letters indicate statistically significant differences (*p* < 0.05). The UVB-L(+) group showed significantly diminished epidermal thickening than the UVB-L(−) group.

**Figure 4 ijms-23-07009-f004:**
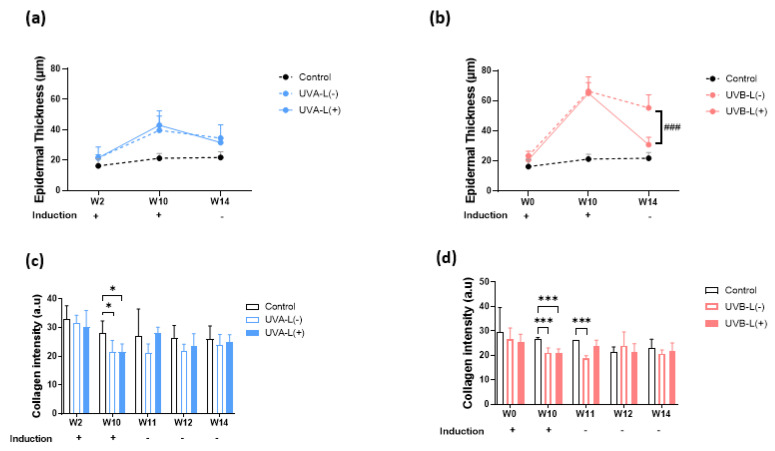
Effect of 755 nm picosecond laser on OCT epidermal thickness and collagen intensity in (**a**,**c**) UVA-irradiated and (**b**,**d**) UVB-irradiated nude mice. W2, UVA baseline; W10, complete induction; W14, post laser treatment. (**a**,**b**) UVB-L(+) group demonstrated significantly accelerated epidermal thickening recovery compared to UVB-L(−) group. There were no significant differences between UVA-L(−) and UVA-L(+) groups in epidermal thickening recovery. (**c**,**d**)The UVA-L(+) group regained more collagen at W14 than UVA-L(−) group, *p* < 0.05; Figure (**c**), whereas no significant difference in collagen intensity was observed in the UVB-L(−) and UVB-L(+) groups at W14. *, *p* < 0.05,***, *p* < 0.001: significant difference vs. the control group. ###, *p* < 0.001: significant difference vs. the induction group.

**Figure 5 ijms-23-07009-f005:**
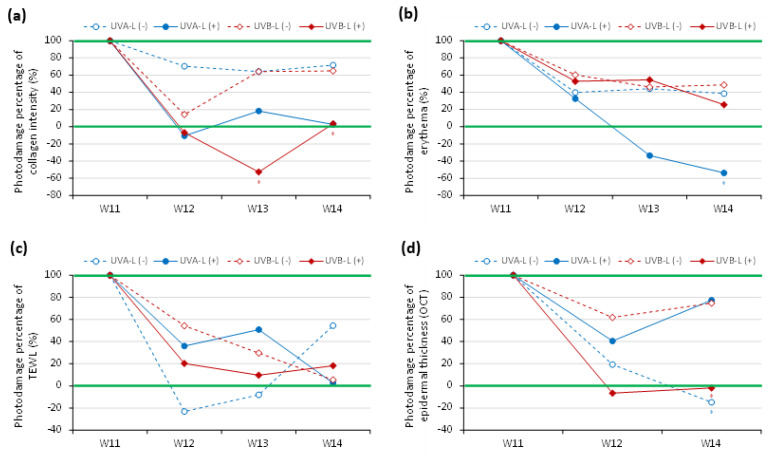
Photodamage percentile curve in UVA-/UVB-irradiated groups with spontaneous reversal (dotted lines) and laser-assisted reversal (solid lines). (**a**) collagen intensity, (**b**) erythema, (**c**) TEWL, and (**d**) epidermal thickening measured by OCT.

**Figure 6 ijms-23-07009-f006:**
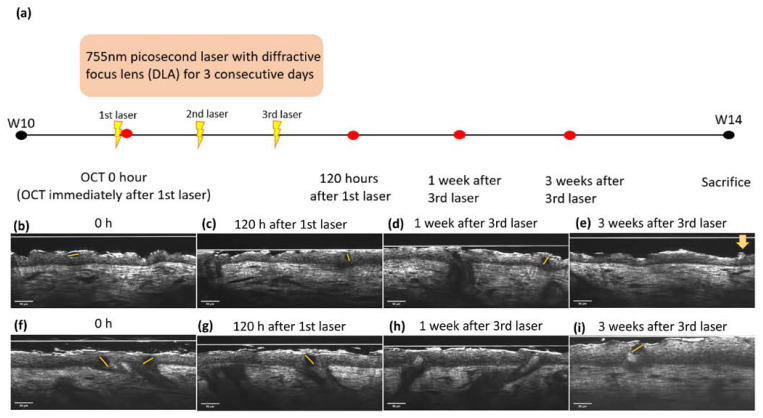
(**a**) Time frame of 755 nm picosecond laser with diffractive lens array (DLA) and OCT scanning for LIOB. LIOB appears as a dark, hypo-reflective vacuole (star) in the epidermis, detected by real-time OCT images. UVA-L(+) group: (**b**) immediately and (**c**) 120 h after the initial laser treatment, (**d**) 1 week, and (**e**) 3 weeks after the third laser treatment. Microscopic epidermal necrotic debris (MENDS) was observed as a hyper-reflective stack immediately inferior to and within the stratum corneum (yellow arrow). UVB-L(+) group: (**f**–**i**). The size of the LIOB in the UVB-L(+) group was larger than that in the UVA-L(+) group. The dotted lines depicted the LIOB diameter.

## Data Availability

The data supporting the findings of this study are available in this article and the Appendix A.

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
