# Peer review of "Photoaging and Sequential Function Reversal with Cellular-Resolution Optical Coherence Tomography in a Nude Mice Model"

_ijms, 2022, doi:10.3390/ijms23137009_

Round 1

Reviewer 1 Report

The manuscript presents a systematic study of the effect of UVA and UVB radiation on skin photoaging by FF OCT method in a murine model. In addition, a piscosecond alexandrite laser was used to induce LIOB in the skin and reverse the photoaging. The effects of the photoaging and laser irradiation on skin morphology was systematically assessed and the significance of the results tested by statistical tests.

The study is well written, the results discussed and their implication for humans explained. It is an interesting study, which complements similar studies performed by ordinary OCT, RCM and histological techniques.

I have a few minor comments:

Results:

-          The parameters in the “Post induction recovery ratio” must be properly induced. Since the equations are already present in Fig. 1, there is no need write them again in the text (l.73).

-          Figures 2 and 3 – Histology images are not included in the caption. Please, add description.

-          L.144: “LIOB was still observed in …” An explanation to the reader, how LIOB in OCT images looks like, is missing in the text.

-          Figure 4: Labels are too small. Markers (* and dotted lines) are missing or too small/thin.

Reviewer 2 Report

The manuscript is aimed to establish a photoaging reversal nude mice model using full-field optical coherence tomography (FFOCT). FFOCT has spatial resolution about 1 μm and acquires cellular details noninvasively.

Comments:

1. The main issue is the conception of this study.  In fact, three experimental tools were used additionally to histology analysis: EFFECT, spectrophotometer (SCM-104/108, Ruyico Technology Corporation, Taipei, Tai-366 wan), and DermaLab Combo® (Cortex Technology, Denmark). In the Result section, FFOCT  was used as the main method in the only subsection from eight in total. Therefore, the results do not correspond to the aim of the paper.  The authors should make efforts to coincide  the aim and the paper content.  

2. The presented OCT images do not look as high-resolutions (1 μm) ones. It will be nice if the authors add the examples of 1 μm resolution OCT images.

3. Can the authors give a description of biological meaning of the term “collagen intensity”. It should be point out that many pathological processes like inflammation, fibrosis, lymphedema cause collagen disorganization.  Had this effect been taken into account?     

4. Lines 72-73. All parameters in formulas should be described here.   

5. English should be improved.  

Round 2

Reviewer 2 Report

Dear authors!

Thank you for your efforts to improve this manuscropt